# Emergence of High Pathogenicity Avian Influenza Virus H5N1 Clade 2.3.4.4b in Wild Birds and Poultry in Botswana

**DOI:** 10.3390/v14122601

**Published:** 2022-11-22

**Authors:** Samantha L. Letsholo, Joe James, Stephanie M. Meyer, Alexander M. P. Byrne, Scott M. Reid, Tirumala B. K. Settypalli, Sneha Datta, Letlhogile Oarabile, Obakeng Kemolatlhe, Kgakgamatso T. Pebe, Bruce R. Mafonko, Tebogo J. Kgotlele, Kago Kumile, Boitumelo Modise, Carter Thanda, John F. C. Nyange, Chandapiwa Marobela-Raborokgwe, Giovanni Cattoli, Charles E. Lamien, Ian H. Brown, William G. Dundon, Ashley C. Banyard

**Affiliations:** 1Botswana National Veterinary Laboratory (BNVL), Private Bag 0035, Gaborone, Botswana; 2Animal and Plant Health Agency (APHA)—Woodham Ln, Addlestone KT15 3NB, UK; 3Animal Production and Health Laboratory (APHL), United Nations Food and Agriculture Organisation (FAO)/International Atomic Energy Agency (IAEA) Agriculture and Biotechnology Laboratory, IAEA Laboratories, Friedenstrasse 1, 2444 Seibersdorf, Austria; 4Department of Veterinary Services (DVS), Ministry of Agriculture, Private Bag 0032, Gaborone, Botswana

**Keywords:** high-pathogenicity avian influenza, fish eagle, poultry, doves, H5N1, high-pathogenicity, mass mortality

## Abstract

Numerous outbreaks of high-pathogenicity avian influenza (HPAI) were reported during 2020–2021. In Africa, H5Nx has been detected in Benin, Burkina Faso, Nigeria, Senegal, Lesotho, Namibia and South Africa in both wild birds and poultry. Botswana reported its first outbreak of HPAI to the World Organisation for Animal Health (WOAH) in 2021. An H5N1 virus was detected in a fish eagle, doves, and chickens. Full genome sequence analysis revealed that the virus belonged to clade 2.3.4.4b and showed high identity within haemagglutinin (HA) and neuraminidase proteins (NA) for viruses identified across a geographically broad range of locations. The detection of H5N1 in Botswana has important implications for disease management, wild bird conservation, tourism, public health, economic empowerment of vulnerable communities and food security in the region.

## 1. Introduction

High-pathogenicity avian influenza (HPAI) constitutes a significant risk to wild bird and poultry health wherever it emerges [1]. The zoonotic risk associated with animal influenza A viruses means that emergence of these viruses at the human–animal interface leads to public health risks of infection. Human infection with H5 avian influenza (AI) viruses has been reported in several countries during the recent epizootics including in Russia and in the United Kingdom [2,3]. Whilst infection of Anseriform species may be asymptomatic, when high-pathogenicity avian influenza virus (HPAIV) infects Galliformes, the outcome invariably results in high levels of morbidity and mass mortality events. Between 2020 and 2022, significant outbreaks of HPAI have occurred across Europe having an unprecedented impact on both wild bird populations and the poultry sector. Furthermore, during this period the virus has spread, most likely through asymptomatic infection of migratory birds, to North America, initially being detected in Canada and later sweeping across the USA [4]. These recent incursions have all been caused by HPAIV subtype H5Nx clade 2.3.4.4. During 2021/22, epizootics have been caused almost exclusively by H5N1 clade 2.3.4.4b. Whilst in developing countries both passive and active surveillance activities enable rapid detection and assessment of such incursions, across resource-limited settings, the detection and characterization of these viruses is less frequent. [5]. Within Africa, H5Nx avian influenza viruses (AIVs) have been detected, predominantly by nucleic acid detection in numerous wild bird species and poultry in Benin, Burkina Faso, Cameroon, Côte d’Ivoire, Ghana, Lesotho, Mali, Mauritania, Namibia, Niger, Nigeria, Senegal, South Africa, and Togo [6]. A diverse range of viral genotypes have been detected including H5N8, H5N6, and H5N1 [7]. The migration of wild birds along the Black Sea/Mediterranean and East African/West Asian migratory bird pathways may have facilitated the emergence of HPAIV H5N1 clade 2.3.4.4 outbreaks across the African continent although a complete understanding of the dynamics of virus infection and bird migration both to and across the African continent requires further evaluation [8,9] The virus was introduced into South Africa in April 2021 and Lesotho in May 2021 [10,11].

Risks associated with the incursion of HPAIV into new areas are significant. The impact on veterinary public health, the poultry industry, and food security of the affected region can be serious [12]. In addition, wild bird species (particularly endangered species) can be severely impacted, and this can have negative implications for other sectors, including tourism [13,14]. Avian influenza incursions can hinder a country’s attempts to diversify the economy and improve the livelihoods of the most vulnerable communities, particularly women in rural areas. Emergency disease investigations were instigated in response to the incursion of HPAIV subtype H5N1 clade 2.3.4.4b as the causative agent of bird mortalities in Botswana. Detailed here are the events that occurred during this period, pathological findings associated with infection, and genetic characterisation of the infecting virus.

## 2. Materials and Methods

### 2.1. Surveillance Activities within Botswana

The poultry industry in Botswana is still in its infancy, with the broiler sector producing a total of 51,588 tons of chicken meat and the layer sector producing 12,747,763 table eggs during the 2020/2021 reporting year [15]. A state of high alert was instituted in Botswana in response to a H5N1 HPAI outbreak during April 2021 in neighboring South Africa. This state of increased alert resulted in the enhancement of active surveillance for HPAI in both domestic and wild birds across Botswana, especially around permanent water bodies. Die-offs of poultry and wild birds were observed between June and September 2021 with mass mortality events involving more than 7000 birds being detected in poultry in Bokaa, Kgatleng District, and around the Okavango Delta, Northwest District. A further mortality event was recorded in African mourning doves in Okavango Delta, Northwest District of Botswana.

In response to the HPAI H5N1 outbreak in the Republic of South Africa during April 2021, surveillance activities by the Department of Veterinary Services (DVS) and Department of Wildlife and National Parks (DNWP) in Botswana intensified, shifting from a mainly passive surveillance approach with a small active surveillance component, to a resource-intensive active surveillance. The initial surveillance was undertaken in ostrich farms around the Mahalapye area and in poultry within a 10 km radius around the Multi-Species Abattoir (MSAB). This then intensified into a countrywide active surveillance and involved surveillance at border posts with South Africa to detect potential illegal introduction of poultry and poultry products, surveying of the national parks for dead and sick birds (especially around water bodies), and active surveillance around both natural and man-made water bodies, especially in areas deemed to be high risk areas for HPAI in Botswana. Through increased public awareness campaigns, sick or dead birds were also reported to the Department of Veterinary Services (DVS) by members of the public. Data was collated through the DVS during the reported period from June–September 2021. Birds submitted were processed as described below.

### 2.2. Pathological Assessment of Submitted Carcasses

Submitted carcasses were subjected to full post-mortem examination (PME) wherever possible at the Botswana National Veterinary Laboratory (BNVL). Where carcasses were unavailable, faecal samples were submitted for influenza A virological testing. Samples collected were stored at −80 °C until further virological analysis.

### 2.3. Virological Investigation

Tissues taken from domestic and wild bird carcasses were assessed for influenza A virus nucleic acid using a matrix (M) gene-specific real-time reverse-transcriptase polymerase chain reaction (rRT-PCR) assay [16] followed by subtype specific rRT-PCRs [17,18,19] to identify the haemagglutinin (HA) gene and Newcastle disease virus specific assay [20] at Botswana National Veterinary Laboratory (BNVL). Duplicates of the tissues were subjected to virus isolation in embryonated specific antibody negative (SAN) chicken eggs as described previously [21]. Tissue smear samples on Whatmann filter paper and extracted RNA were sent to international collaborators for further evaluation (APHA and APHL). Confirmatory testing was undertaken using M gene specific [‘Nagy2’] and subtype-specific rRT-PCRs [19,22,23] to define both HA and neuraminidase (NA subtypes). An HPAIV H5-specific detection PCR was also used to rapidly confirm the presence of a multi-basic HP cleavage site as described [24]. Virus isolation was attempted [25,26] and where successful, whole-genome sequence (WGS) data was generated using an Illumina MiSeq (Illumina inc., San Diego, USA), as described previously [27]. Raw sequencing reads were assembled using a custom script, denovoAssembly (https://github.com/AMPByrne/WGS/blob/master/denovoAssembly_Public.sh accessed on 7 July 2022). Sequences generated were compared to contemporary H5 2.3.4.4b viruses downloaded from the GISAID EpiFlu database (https://platform.gisaid.org accessed on 9 August 2021). Gene sequences were then aligned using Mafft v7.487 [28] and manually trimmed to the open-reading frame using AliView [29]. Phylogenetic trees were then inferred using the maximum-likelihood approach in IQ-Tree v2.1.4 [30] with ModelFinder [31] to infer the appropriate phylogenetic model and 1000 ultrafast bootstraps [32]. Phylogenetics trees were visualised and annotated using FigTree v1.4.4. All sequences generated in the study are available through the GISAID EpiFlu Database: EPI_ISL_12045918 (A/Dove/Botswana/2097/2021), EPI_ISL_12045319 (A/chicken/Botswana/2163-B/2021), EPI_ISL_12045320 (A/chicken/Botswana/2248/2021), EPI_ISL_12045920 (A/fish eagle/Botswana/1338/2021), and EPI_ISL_12045919(A/Dove/Botswana/2334/2021).

## 3. Results and Discussion

The threat of viral incursion from neighboring countries drove nationwide surveillance activities for AI in Botswana. Between 3 June and 13 September 2021, many wild and domestic bird carcasses were found across the country (Appendix A). Surveillance activities targeted high-risk areas including regions containing large water bodies and areas with a high poultry density, especially back yard poultry. Wild and domestic bird populations were surveyed both actively and passively by the farming community and wildlife and veterinary officers. Surveillance activities targeted sick, dying, or dead birds and faecal material from wild birds around large permanent water bodies, farms with established wild bird linkages, and farms containing potential bridging species such as Columbiformes. Samples collected for assessment are detailed in Appendix A.

A timeline of the observed mortalities and geographical distance of the sites where the dead or sick birds were found, likely indicated independent incursions of disease as infected individuals would not be expected to move long distances between colonies (Figure 1). There was a cluster of wild and domestic cases in the northwest of Botswana and a single case in the southeast of Botswana, Kgatleng District, which was highly suggestive of at least two different introductions of the virus into the country. Human translocation of infected birds/material from infected countries was considered possible especially because the village is located near the border with South Africa, a country which had an active HPAIV H5N1 clade 2.3.3.4b outbreak. However, the possible introduction of the disease from the neighbouring country either through wild birds or human translocation could not be tested due to a lack of publicly available sequence data from South Africa from the time of the outbreak. Clinical signs observed in different species included listlessness, drooping wings, incoordination, paralysis, and convulsions signifying neurological impairment. Other observed clinical signs included inappetence, diarrhea and respiratory signs (dyspnea, nasal, and ocular discharge), and periorbital oedema. Birds succumbed to infection within 24–36 h of the development of clinical disease. In contrast, infection of Galliformes was characterized by sudden death, preceded by no clinical signs. A total of 7282 domestic birds died during the course of the outbreak, 6000 of which were broilers from one farm. Of these, 248 birds across 6 recorded species were submitted for assessment (Appendix A).

A total of two hundred and forty-eight carcasses were examined by BNVL [African fish eagle (*n* = 1), African mourning doves (*n* = 7), helmeted guinea fowl (*n* = 3), duck (*n* = 4), goose (*n* = 1) and chicken (*n* = 233)]. Of the birds examined, twenty-five were associated with HPAIV H5N1 rRT-PCR positive cases and an isolate was obtained from only one of these cases using virus isolation. All samples were Newcastle disease negative both on rRT-PCR and on virus isolation. The majority of the domestic birds were in fair to good body condition with one dove (BNVL 2021/2365) having a poor body condition. An empty proventriculus and gizzard was noted in the African fish eagle at PME. The most prominent findings associated with the HPAIV H5N1 infection were dehydrated carcasses, congested heart, and oedematous brain (Figure 2). Approximately 76% of chickens, 100% of African fish eagle (dehydration and oedematous brain, alone), and 100% of African mourning doves infected with H5N1 presented with these findings. Congested and oedematous lung, distended congested intestines and pale kidneys were observed in approximately 60% of the birds. Distended intestines and pale kidneys were noted in the chickens (78%) only while both doves (100%) and thirteen chickens (72%) had a congested and oedematous lung. Other noted lesions in the wild birds included sunken eyes, soiled ventral feathers, and icteric heart, liver, and spleen. Only one dove had moderate swelling with slight icterus. Generalized subcutaneous congestion, congested ovarian follicles, pale fatty liver and spleen, and cyanotic comb, beak and hocks were the least frequently observed clinical signs and observed in chickens only with a frequency of 4% (Figure 2).

African fish eagles are a monogamous large non-migratory eagle species that are found near large open water bodies in Sub-Saharan Africa including Botswana with a population size of approximately 300,000 birds [33,34]. These carnivorous birds are diurnal and are very effective hunters that prey mainly on fish and occasionally on other birds [33,34]. Waterbirds often targeted by the African fish eagles include ducks, ibis, storks, herons, and greater and lesser flamingos [33]. These are well known for their kleptoparasitism, often stealing food from other bird species such as the goliath herons [33]. Adult African fish eagles are territorial birds which live in pairs [33]. During enhanced surveillance, two African fish eagles were found, one dead and decomposing and the other sick, succumbing shortly after discovery. The detection of infection in African fish eagles was likely the result of their status as apex predators and their piratical behaviours.

African mourning doves are a monogamous (seasonal or lifelong) small non-migratory bird species that have an extremely large range associated with moist savanna, cultivated areas, water bodies and riverine Acacia woodlands in Sub-Saharan Africa [34,35]. This species is a native resident in southern Africa, including northern Namibia, northern Botswana, southern and northern Zimbabwe, Mozambique, and north-eastern South Africa [34]. In Botswana, these birds are found almost exclusively in the Okavango Delta area [34]. Their population size is unknown despite the species being one of the most common in Africa [35]. African mourning doves are omnivorous birds feeding mainly on seeds and grain and occasionally on termites [34] and as such, their feeding behaviours bring them into direct contact with chicken housing where grain is used. They co-exist well with other birds in their habitat, are non-migratory, live in solitude or in pairs and have a lifespan of five to six years [35]. Flocks have been known to occur, especially in habitats around water bodies [35]. The infected doves detected in this study may therefore have become infected through interaction with Galliformes or Anseriformes, through chicken flocks or wild waterfowl, respectively.

The genome sequences consisting of at least the HA gene were obtained from five samples across five cases (Table 1). Genetic assessment of the virus infecting the wild and domestic birds established the relationship of the H5N1 HPAIV involved with the sequences of previous H5N1 viruses detected across Europe and Africa during 2020/21 (Figure 3). All genes clustered closely with isolates detected during the 2020/21 season including sequences from Europe and Africa (Senegal, Benin, Lesotho, Burkina Faso). However, the sequences from Botswana do appear to form a distinct sub-clade, demonstrating higher relatedness to each other than with other sequences from the same time period (Figure 3 and Appendix A). This is as expected given the relatively close geographic and temporal relationship between most of the detections in Botswana (Figure 1). The cleavage site (CS) sequence motif for the wild and domestic bird isolates was PLREKRRKRGLF, as reported across the vast majority of HPAIV CS sequences determined over 2020/21 across Africa, the United Kingdom and continental Europe.

Apart from the detection of dead or diseased birds from domestic poultry holdings associated with these cases, no further mass mortalities in resident bird species were observed during the reported outbreak period. Considering that the wildlife bird species affected in the northwest include doves and an African fish eagle, and that the index case was the African fish eagle, there was a very high probability that the disease was introduced through migratory birds. Especially since the viruses have a high similarity with viruses from West Africa and Europe from the 2020/21 autumn/winter season, and which include migratory bird species such as the White Stork [36]. The cases were in various areas surrounding a large permanent water body that harbors migratory birds including flamingos, geese, ducks, and vultures. Interestingly, none of the migratory birds were found exhibiting clinical disease during the enhanced surveillance period. However, most of the cases, including the index case, occurred in areas of poor accessibility, hence there is a possibility that there were sick or dead wild birds that went unnoticed in this area. The significance of the index case in the northwest of Botswana being an African fish eagle is that this species is non-migratory and do not move very far from the waterbody they nest near. As such, the infection of this species, especially in a country that has no previous history of HPAI and in the middle of a national park, most likely indicates that the HPAI originated from an unidentified migratory bird(s). The origin of the case from backyard chickens in Bokaa, Orpington, was unidentified and may have been due to contaminated fomites or a bridging wildlife species like a dove sharing the watering or feeding points with the chickens.

## 4. Conclusions

The detection of sequential deaths including mass mortality events in Botswana demonstrates the potential threat of infectious diseases to bird diversity in the Okavango Delta and to the fledgling poultry industry in Botswana. There remains uncertainty as to whether the mortalities across multiple locations resulted following independent incursions or whether local wild bird movements led to introduction or onward spread within the country. The species of bird that may have introduced the virus remains unclear with active surveillance targeting bird populations being the only way to ascertain species that carry and shed the virus in the absence of clinical disease. Wild birds are included in the 3-year National Disease Surveillance Plan; however, the inaccessibility of the region also leads to the possibility that further unidentified victims of the infection may exist. The impact of the HPAI epizootic on both the wild bird population and poultry industry were limited, likely due to the rapid response of the DVS and DWNP to the outbreaks and the absence of densely populated poultry premises in the country. The consequences of these outbreaks to the wildlife population and the breeding season of the African mourning dove are likely to be insignificant where populations are strong. However, the risk of this virus to other species that are threatened within the region and across neighboring areas may be more severe. Future census activities will help to define where losses have impacted the wild bird species in the Okavango Delta, especially the 25 threatened species that reside at the Delta [37]. Gaining a better understanding of these outbreaks is critical to understanding the transmission dynamics of HPAIV in Botswana and the potential for further geographical spread. Knowledge of this virus and the susceptibility of different species will enable longer term planning for poultry sector development and the protection of endangered species.

## Figures and Tables

**Figure 1 viruses-14-02601-f001:**
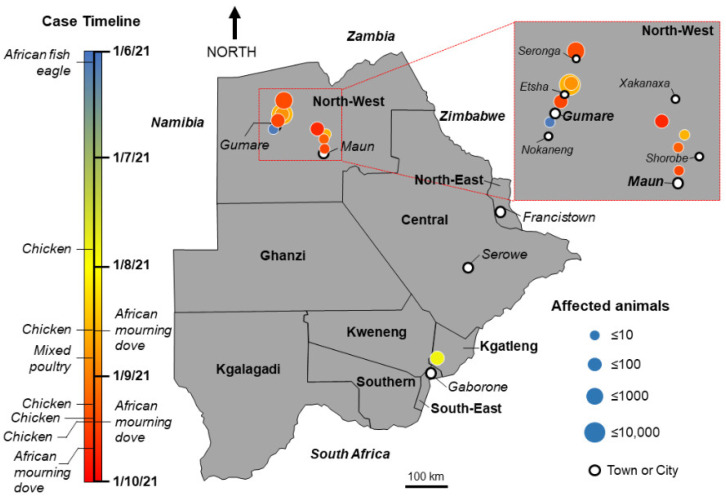
Samples submitted for diagnostic assessment and geographical distribution of domestic and wild bird carcasses including timeline of detection. The distribution of observed deaths is shown on the map. Cases are coloured according to time of observation as denoted in the key. The size of circle is proportional to the number of cases with larger circles denoting a greater number of observed mortalities as per the key. Positive cases are also noted.

**Figure 2 viruses-14-02601-f002:**
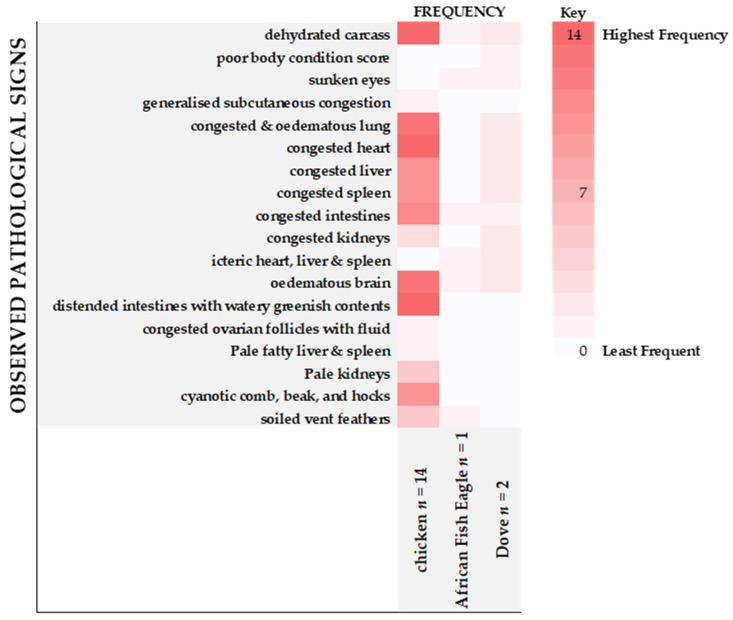
Gross pathological findings. Gross pathological lesions observed from the birds submitted for necropsy and later confirmed for HPAIV H5N1 during winter/spring of 2021. Main gross pathological findings included cyanosis around the hocks and combs. Changes in the abdominal cavity were observed primarily in the liver, spleen and intestines with congestion and oedema being observed. Species examined included chicken (*Gallus gallus domesticus*), African fish eagle (*Haliaeetus vocifer*), and African mourning dove (*Streptopelia semitorquata*). The number of carcasses assessed for each species is indicated at the bottom of the chart. Numbers captured in the chart refer to the number of birds that presented with the lesion. The colour range in the key reflects the frequency of the pathological sign observed from least (light) to highest (dark) frequency.

**Figure 3 viruses-14-02601-f003:**
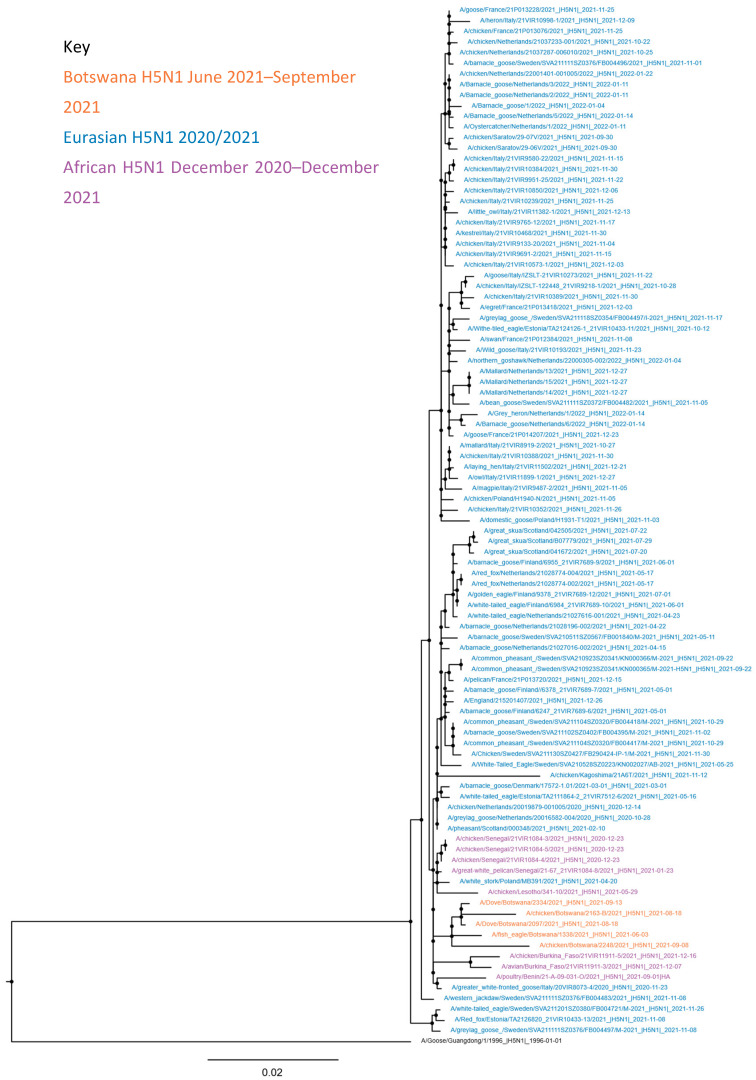
Maximum-likelihood phylogenetic tree of the HA gene segment. As per the key, H5N1 sequences detected in Botswana between July and September 2021 are highlighted in orange, Eurasian H5N1 sequences detected during 2020 and 2021 are highlighted in blue, and H5N1 sequences detected in Africa between December 2020 and December 2021 are highlighted in purple. Relationships among the Eurasian and African 2020/21 H5 HPAIV strains were inferred by adding the novel sequence data from Botswana to that available on GISAID Epiflu on 25 February 2022. The sequences were then filtered to include only Influenza A virus subtype H5N1 and are rooted to A/Goose/Guangdong/1/1996_|H5N1|_1996-01-01. In September 2022, further H5N1 sequences were added from GISAID for Burkina Faso and from GENBANK for Benin and Lesotho. Sequences were aligned using MAFFT v7.407 and phylogenetic tree inferred using Miyawaki with ultrafast bootstrap node support. Trees were visualised in FigTree v1.4.4.

**Table 1 viruses-14-02601-t001:** Overview of samples that tested positive for H5N1 HPAIV and full genome sequence was determined.

Sample Location (and Collection Date)	Species	Sample ID	Sample Type	Accession Number
Habu, Okavango Delta 3 June 2021	African fish eagle (*Haliaeetus vocifer*)	BNVL 2021/1338	Pooled tissues (lung, trachea, liver, Spleen, intestine, proventriculus)	EPI_ISL_12045920
Sedie, Maun East 18 August 2021	African mourning dove (*Streptopelia decipiens*)	BNVL 2021/2097	Pooled tissues (lung, trachea, liver, spleen, intestine, proventriculus)	EPI_ISL_12045918
Etsha 13, Gumare Sub-District	Broiler backyard chickens 18 August 2021	BNVL 2021/2163	Trachea	EPI_ISL_12045319
	African mourning dove (*Streptopelia decipiens*) 13 September 2021	BNVL 2021/2334	Pooled tissues (Lung, trachea, liver, spleen, intestine, proventriculus)	EPI_ISL_12045919
Shorobe, 8 September 2021	Tswana backyard chicken	BNVL 2021/2248	Proventriculus	EPI_ISL_12045320

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
