# Peer review of "Emergence of High Pathogenicity Avian Influenza Virus H5N1 Clade 2.3.4.4b in Wild Birds and Poultry in Botswana"

_viruses, 2022, doi:10.3390/v14122601_

Round 1

Reviewer 1 Report

The manuscript entitled " Emergence of high pathogenicity avian influenza virus H5N1 clade 2.3.4.4b in wild birds and poultry in Botswana” in which the authors have investigated the phylogenetic and genomic aspects of the avian influenza virus H5N1 subtype  among domestic and wild bird species in Botswana  followed by analyzing the complete sequencing of five isolated viruses virus. The study is well-oriented, and the findings are relevant to the field of research. However, a few questions need to be answered before considering the manuscript for publication.

In post-mortem findings, are there any changes in the pancreas, Hemorrhagic skin?

Is there any chance to add figures of clinical and gross pathological findings?

Is there any chance of mixed infection with different subtypes especially in wild and migratory birds?

Why were the authors restricted to five complete genome sequencing? Virus isolation was done on all samples, please clarify. Are there any limitations? Especially when you have feces samples.

It is better to consider having a dedicated discussion section.

Author Response

Letsholo et al., 2022: AIV in Botswana Rebuttal

We thank the reviewer for their constructive comments that have helped to improve the quality of the manuscript.

Reviewer 1 comments:

The manuscript entitled " Emergence of high pathogenicity avian influenza virus H5N1 clade 2.3.4.4b in wild birds and poultry in Botswana” in which the authors have investigated the phylogenetic and genomic aspects of the avian influenza virus H5N1 subtype  among domestic and wild bird species in Botswana  followed by analyzing the complete sequencing of five isolated viruses virus. The study is well-oriented, and the findings are relevant to the field of research. However, a few questions need to be answered before considering the manuscript for publication.

In post-mortem findings, are there any changes in the pancreas, Hemorrhagic skin?

There was no report of haemorrhages under the skin, the main finding with respect to the skin was cyanosis especially around the hocks and combs. The changes in the abdominal cavity were captured mainly with respect to the liver, spleen and intestines as congested and oedematous.  No mention was made of the pancreas itself, mostly the intestines. (Added to Figure 2 legend (yellow)

Is there any chance to add figures of clinical and gross pathological findings?

There were a few pictures taken of clinical findings but they were not very graphic and none of gross pathology. Unfortunately, none were of a standard required for publication

Is there any chance of mixed infection with different subtypes especially in wild and migratory birds?

This is considered unlikely. Descriptions of dual infection is very rarely reported and WGS of samples has not detected HA or NA of other subtypes in the samples presented here or in over 300 full genomes generated in the UK as part of out outbreak response.  So whilst biologically coinfection is possible, there is no evidence for it in the samples presented.

Why were the authors restricted to five complete genome sequencing? Virus isolation was done on all samples, please clarify. Are there any limitations? Especially when you have feces samples.

Virus isolation was undertaken all samples at BNVL but was only successful for a single isolate.  A significant limitation was the provision of specific antibody negative (SAN) egg availability at BNVL.  Faecal samples were used for live wild bird samples around extensive waterbodies, to ensure that we still were surveying these populations despite the challenge of capturing them. Samples shared with colleagues in the UK and Austria were sent of FTA cards as described.

It is better to consider having a dedicated discussion section.

We have followed the template supplied by the journal.

Reviewer 2 Report

This manuscript describes the detection of highly pathogenic H5N1 of the clade 2.3.4.4b sub-lineage in Botswana, for the first time. The disease in a variety of wild bird species and domestic chickens was diagnosed by macroscopic post-mortem evaluation and RT-PCR subtyping, and the genetic sequences for five virus isolates were used for phylogenetic analysis. The novel sequence data from this study will contribute towards ongoing efforts to understand the ecology of HPAI in the African sub-continent. 

I have the following minor or major comments that should be addressed: 

Introduction.

Line 41: define “HPAIV”

Line 46: please add the missing references

Lines 55-58: The timing of the southward migrations of birds (mainly shorebirds) using the Black Sea/Mediterranean and East African/West Asian migratory bird flyways (i.e., in the southern hemisphere spring) are inconsistent with the timing of the 2021 H5 HPAI outbreaks in Botswana and elsewhere in southern Africa (winter), and there is no evidence that these flyways play any role in the dissemination of H5 HPAI “across the African continent” (lines 57-58). There is greater scientific evidence for the role of Palearctic-breeding ducks overwintering in West and Central Africa as carriers of the virus into the continent, and the subsequent intra-continental dissemination by Afrotropical ducks moving in response to summer rainfall patterns. See Khomenko et al., 2018 accessible at https://www.fao.org/3/ca1209en/CA1209EN.pdf and Fusaro et al., 2019, doi: 10.1038/s41467-019-13287-y., and revise accordingly.

 Materials and Methods.

Lines 79-83: were other pathogens like Newcastle disease virus ruled out by laboratory tests as an alternative cause of the mass die-offs, especially in the doves and chickens? Please specify.

Lines 100-103: please specify the laboratory/ies where the post-mortem examinations were conducted.

Lines 109-110: were the eggs used for isolation in Botswana from an SPF flock, or an SAN flock?  Please provide details.

Results and Discussion

Please provide the following data currently missing from the manuscript:

How many viruses were successfully isolated in eggs at either BNVL and/or APHA/APHL?

For which isolates/ samples were complete genomes recovered, and for which were only partial genomes recovered, and in that case, for which genome segments?

Were phylogenetic trees reconstructed for all 8 genome segments, if not why not and if so, please provide the trees as supplemental data?

Lines 151-154: it is a great pity that no sequences from South Africa were included in the phylogenetic analysis; was anyone from South Africa approached to collaborate and provide sequence data at all? You might be surprised at the willingness of scientists to share data, if asked.

Lines 154-162: the clinical signs described are not pathognomonic for HPAI (ditto for Table 1). Were any other laboratory tests conducted to rule out NDV as the cause of the clinical signs and deaths?
Please discuss this in the text.

Table 1 should rather be moved to a supplemental file. It is unclear whether the samples in Table 1 were all tested for the presence of the virus. Please specify whether the samples that are not “positive” were tested and found to be IAV-negative, or whether they were submitted but not tested for IAV.

Lines 172-174- move the information in [  ] to the end of the sentence

Line 174: “Of the birds examined, twenty-five were associated with HPAIV H5N1 positive cases”- please clarify: did these 25 birds test IAV positive using RT-PCR, or was the diagnosis only made on macroscopic pathology?

Figure 2- please neaten the figure, “Post-mortem Findings” heading vs “Gross pathology findings” in the caption; mixed cases in the list of clinical signs.

Line 234: please see the previous comment- how many complete genome sequences were recovered, please include the trees for the other segments.

Conclusions

Line 276- multiple bird species could’ve introduced the viruses

Lines 286-288: is any active surveillance planned in Botswana for 2023, particularly wild birds?

Author Response

Letsholo et al., 2022: AIV in Botswana Rebuttal

We thank the reviewer for their constructive comments that have helped to improve the quality of the manuscript.

Reviewer 2:

This manuscript describes the detection of highly pathogenic H5N1 of the clade 2.3.4.4b sub-lineage in Botswana, for the first time. The disease in a variety of wild bird species and domestic chickens was diagnosed by macroscopic post-mortem evaluation and RT-PCR subtyping, and the genetic sequences for five virus isolates were used for phylogenetic analysis. The novel sequence data from this study will contribute towards ongoing efforts to understand the ecology of HPAI in the African sub-continent. 

I have the following minor or major comments that should be addressed: 

Introduction.

Line 41: define “HPAIV”

Manuscript amended as suggested

Line 46: please add the missing references

Manuscript amended as suggested

Lines 55-58: The timing of the southward migrations of birds (mainly shorebirds) using the Black Sea/Mediterranean and East African/West Asian migratory bird flyways (i.e., in the southern hemisphere spring) are inconsistent with the timing of the 2021 H5 HPAI outbreaks in Botswana and elsewhere in southern Africa (winter), and there is no evidence that these flyways play any role in the dissemination of H5 HPAI “across the African continent” (lines 57-58). There is greater scientific evidence for the role of Palearctic-breeding ducks overwintering in West and Central Africa as carriers of the virus into the continent, and the subsequent intra-continental dissemination by Afrotropical ducks moving in response to summer rainfall patterns. See Khomenko et al., 2018 accessible at https://www.fao.org/3/ca1209en/CA1209EN.pdf and Fusaro et al., 2019, doi: 10.1038/s41467-019-13287-y., and revise accordingly.

We have amended the manuscript to reflect this suggestion and have cited the proposed manuscripts.

Materials and Methods.

Lines 79-83: were other pathogens like Newcastle disease virus ruled out by laboratory tests as an alternative cause of the mass die-offs, especially in the doves and chickens? Please specify.

To rule out NDV infection as a potential causative agent, molecular analysis was undertaken simultaneously for Newcastle disease virus. All samples were negative.

Lines 100-103: please specify the laboratory/ies where the post-mortem examinations were conducted.

Manuscript amended as suggested

Lines 109-110: were the eggs used for isolation in Botswana from an SPF flock, or an SAN flock?  Please provide details.

Specific antibody negative embryonated chicken eggs were used for virus isolation in Botswana.  The flock is procured as specific pathogen free chickens from South Africa but are kept as specific antibody negative chickens in Botswana.

Results and Discussion

Please provide the following data currently missing from the manuscript:

How many viruses were successfully isolated in eggs at either BNVL and/or APHA/APHL?

One virus was successfully isolated from tissues at BNVL while none where at APHA as samples were sent as dried tissue smears on Whatmann FTA cards. Manuscript amended as suggested.

For which isolates/ samples were complete genomes recovered, and for which were only partial genomes recovered, and in that case, for which genome segments?

Full genome sequence was generated for the samples listed in table 2. 

Were phylogenetic trees reconstructed for all 8 genome segments, if not why not and if so, please provide the trees as supplemental data?

SI data was submitted with trees for all segments

Lines 151-154: it is a great pity that no sequences from South Africa were included in the phylogenetic analysis; was anyone from South Africa approached to collaborate and provide sequence data at all? You might be surprised at the willingness of scientists to share data, if asked.

Due to time pressure from the ongoing UK H5N1 epizootic the analysis was limited to Botswanan samples. However, APHA links closely with colleagues in South Africa and will look into linkages to drive collaboration in that area in future. At the time of initial writing no formal notification of H5N1 in South Africa was evident.

Lines 154-162: the clinical signs described are not pathognomonic for HPAI (ditto for Table 1). Were any other laboratory tests conducted to rule out NDV as the cause of the clinical signs and deaths?
Please discuss this in the text.

All the bird samples were subjected to both avian influenza and Newcastle disease rRT-PCR testing and virus isolation and HA/ HI screening for Newcastle disease and avian influenza where a HA positive sample was found. The manuscript has been amended to reflect this.

Table 1 should rather be moved to a supplemental file. It is unclear whether the samples in Table 1 were all tested for the presence of the virus. Please specify whether the samples that are not “positive” were tested and found to be IAV-negative, or whether they were submitted but not tested for IAV.

All the samples in table 1 were tested for AIV and, as stated, the majority were found to be negative. Amendments have been captured in the manuscript per your suggestion the table has been moved to a SI table

Lines 172-174- move the information in [  ] to the end of the sentence

Manuscript amended as suggested

Line 174: “Of the birds examined, twenty-five were associated with HPAIV H5N1 positive cases”- please clarify: did these 25 birds test IAV positive using RT-PCR, or was the diagnosis only made on macroscopic pathology?

They tested positive on rRT-PCR. Manuscript amended to reflect this.

Figure 2- please neaten the figure, “Post-mortem Findings” heading vs “Gross pathology findings” in the caption; mixed cases in the list of clinical signs.

Manuscript amended to reflect this.

Line 234: please see the previous comment- how many complete genome sequences were recovered, please include the trees for the other segments.

The trees for all segments were in the original submission under supplementary figure 1.

Conclusions

Line 276- multiple bird species could’ve introduced the viruses

Noted and manuscript amended to reflect this

Lines 286-288: is any active surveillance planned in Botswana for 2023, particularly wild birds?

There is active avian influenza/ Newcastle disease surveillance conducted annually that includes assessment of wild birds. This surveillance has been conducted in 2022 and is planned for implementation as part of the current national disease surveillance plan between 2022 – 2025.

Round 2

Reviewer 2 Report

Thank you, all suggested changes have been made.